# Endometrial Cancer and BRCA Mutations: A Systematic Review

**DOI:** 10.3390/jcm11113114

**Published:** 2022-05-31

**Authors:** Maria Luisa Gasparri, Serena Bellaminutti, Ammad Ahmad Farooqi, Ilaria Cuccu, Violante Di Donato, Andrea Papadia

**Affiliations:** 1Department of Gynecology and Obstetrics, Ente Ospedaliero Cantonale, Ospedale Regionale di Lugano, Via Tesserete 46, 6900 Lugano, Switzerland; marialuisa.gasparri@eoc.ch (M.L.G.); andrea.papadia@eoc.ch (A.P.); 2Faculty of Biomedical Sciences, Università della Svizzera Italiana, Via Giuseppe Buffi 13, 6900 Lugano, Switzerland; 3Gynecology Unit, Centro Medico Lugano, Via Petrini 2, 6900 Lugano, Switzerland; 4Gynecology and Fertility Unit, Procrea Institute, Via Clemente Maraini 8, 6900 Lugano, Switzerland; 5Institute of Biomedical and Genetic Engineering (IBGE), Islamabad 54000, Pakistan; farooqiammadahmad@gmail.com; 6Department of Maternal and Child Health and Urological Sciences, Sapienza University of Rome, Policlinico Umberto I, Viale del Policlinico 166, 00161 Rome, Italy; ilaria.cuccu@uniroma1.it (I.C.); violante.didonato@uniroma1.it (V.D.D.)

**Keywords:** endometrial cancer, BRCA1, BRCA2, risk-reducing salpingo-oophorectomy, hysterectomy

## Abstract

This systematic review identifies, evaluates, and summarises the findings of all relevant individual studies on the prevalence of BRCA mutation (BRCAm) in endometrial cancer patients and the incidence of endometrial cancer in BRCAm women patients. Consequently, the benefits and limits of a prophylactic hysterectomy at the time of the risk-reducing salpingo-oophorectomy are analysed and discussed. A systematic literature search was performed in the databases of PubMed, Cochrane, and Web of Science until May 2022; 13 studies met the eligibility criteria. Overall, 1613 endometrial cancer patients from 11 cohorts were tested for BRCA1/2 mutation. BRCA1/2m were identified in 4.3% of women with endometrial cancer (70/1613). BRCA1m was the most represented (71.4%) pathogenic variant. Alongside, a total of 209 BRCAm carriers from 14 studies were diagnosed with endometrial cancer. Only 5 out of 14 studies found a correlation between BRCAm and an increased risk of endometrial cancer. Nevertheless, two studies found a statistical difference only for BRCA1m women. The present systematic review does not provide strong evidence in favour of performing routine hysterectomy at the time of risk-reducing salpingo-oophorectomy; however, it provides epidemiological data that can be useful for counselling patients in order to offer a tailored approach.

## 1. Introduction

Breast cancer gene 1, located on the long arm (q) of chromosome 17, and the BRCA 2 gene, located on the long arm of chromosome 13, are both autosomal dominant tumour suppressor genes involved in DNA damage repair before cell replication. The lifetime risk of breast and ovarian cancer increases for those carrying a pathogenic variant of breast cancer gene 1 (BRCA1) or breast cancer gene 2 (BRCA2) by 40–80% and 11–40%, respectively [1]. In order to reduce the lifetime risk of breast, ovarian, and fallopian tube cancer, NCCN guidelines (National Comprehensive Cancer Network) currently consider a risk-reducing mastectomy (RRM) and recommend salpingo-oophorectomy (RRSO) in women with BRCA mutations (pathogenic variants) (BRCAm) [2]. RRSO is associated with a 42% and 94% reduced risk of developing breast and ovarian cancer in BRCAm carriers, respectively [3], and a 60% reduced all-cause mortality [4].

BRCAm carriers are exposed to a higher risk of other less frequent cancers such as fallopian tube cancer and primary peritoneal cancer [5], pancreatic cancer [6], prostate, and gastric cancer [7].

The risk of uterine cancer in BRCAm women and, consequently, its role as part of the BRCA mutated syndrome is still debated. The analogies between uterine, mainly serous carcinoma and serous ovarian carcinoma, have led to the investigation of potential common pathogenetic features, as well as hereditary causes. Positive family history is noted in approximately 10% of cases of endometrial cancer, suggesting an inherited predisposition, even if the precise genes pattern involved are largely unknown [8].

Uterine serous carcinoma, representing less than 10% of all endometrial cancers, is an aggressive histologic subtype with a poor prognosis. It accounts for about 25% of the entire endometrial cancer mortality, with an overall survival rate at 5 years of 18–27% due to frequent advanced disease at diagnosis and a high rate of distant recurrences even in patients with early-stage disease [9,10]

Some authors [11,12,13] confirmed a higher risk of endometrial cancer in BRCAm women, especially for uterine serous cancer in BRCA1m, while others [14,15] did not support this correlation.

Hence, while RRSO is a well-established procedure for women with BRCAm at 35–40 years of age for BRCA1m and at 40–45 years for BRCA2m [2], prophylactic hysterectomy at the time of RRSO is still a matter of debate [16]. If women carrying BRCA1m are confirmed to be at an increased risk for serous or serous-like endometrial cancer, this should be considered when counselling a patient with regards to the risks and benefits of the addition of hysterectomy at the time of RRSO.

This systematic review aims to assess the risk of endometrial cancer and examine the benefits and limits of a prophylactic hysterectomy in BRCAm women, analysing both the prevalence of BRCAm in endometrial cancer patients and the incidence of endometrial cancer found in BRCAm patients undergoing RRSO with hysterectomy.

## 2. Material and Methods

### 2.1. Information Sources

A systematic literature search was performed in the databases of PubMed, Cochrane, and Web of Science until May 2022. No beginning date limit or language restrictions were used. The review was conducted following the Preferred Reporting Items for Systematic Reviews and Meta-Analyses (PRISMA) statement [17].

### 2.2. Search Strategy

The search terms consisted of “BRCA” and “endometrial cancer” or “uterine cancer”. Reference lists of identified systematic reviews and included studies were manually screened for any other eligible studies.

### 2.3. Study Selection

Titles and abstracts were screened. Articles reporting the incidence of endometrial cancer in BRCAm women or the prevalence of BRCAm in patients affected by endometrial cancer were obtained in full for further evaluation. Studies were excluded if they were case reports, editorials, reviews, or short communications because they did not provide sufficient information to assess the methodological quality.

Title and abstract screening, as well as full-text screening, were performed independently and simultaneously by two authors (SB and MLG) based on pre-defined criteria. All dissents were resolved by consensus.

### 2.4. Data Extraction

For each eligible article, information was collected concerning the first author, year of publication, country of origin, study period, design of the study, the total number of patients, mean or median age, genotyping testing method (including different BRCA deletions and other genes investigated), number of endometrial cancers and uterine serous carcinoma, FIGO stage, previous Tamoxifen use, history of breast cancer, and type of BRCAm. Median follow-up was expressed in years or women-years. Standard Incidence Ratios (SIR) were reported for assessing endometrial cancer risk in BRCAm women when available.

### 2.5. Quality Assessment

The evaluation of the risk of bias in estimates of the comparative effectiveness (harm or benefit) of interventions from the included studies was performed with the “Risk Of Bias In Non-randomized Studies-of Interventions” (ROBINS-I) tool [18].

## 3. Results

### 3.1. Literature Search

Overall, a total of 291 records were identified. After removing 77 duplicates, 214 manuscripts were screened, and 165 were excluded based on the abstract. A full text was obtained for 48 of 49 records. At the end of the screening process, 24 full-text articles were included in the systematic review. All papers were in English.

Titles/abstracts were screened according to the inclusion and exclusion criteria. Most manuscripts were excluded during the screening process due to differing study objectives (*n* = 13), publication types such as editorial or review (*n* = 8), and incomplete data (*n* = 2).

Details about the literature search results are reported in Figure 1. A total of 11/25 included studies that assessed the prevalence of BRCAm in patients affected by endometrial cancer [8,19,20,21,22,23,24,25,26,27,28], and 14/25 included studies that evaluated the incidence of endometrial cancer in BRCAm [11,12,13,14,15,29,30,31,32,33,34,35,36,37].

### 3.2. Patients Characteristics of the Included Studies

Overall, the total number of patients analysed in this systematic review was 37,286.

The number of patients ranged between 20 and 628 in the included studies concerning the incidence of BRCAm in patients with endometrial cancer and between 315 and 14,621 in the included studies concerning the incidence of endometrial cancer in BRCAm patients.

The mean/median age of patients ranged between 20 and 72 years; most of the included patients had a FIGO stage I, and median follow-up ranged between 1.5–9 years and 1.779–59.199 women-years.

### 3.3. Methodological Aspects of the Included Studies

A total of 13 observational retrospective cohort studies, 3 retrospective case-control studies, 7 observational prospective cohort studies, 1 prospective case-control study, and 1 longitudinal cohort study were included in this systematic review. Of these, 10 were multicenter-based studies. Additionally, four studies included only Jewish women, and one study included only patients with hereditary endometrial cancer (Lynch syndrome and hereditary breast-ovarian cancer). The main characteristics of eligible studies on the prevalence of BRCAm in patients with endometrial cancer and on the incidence of endometrial cancer in BRCAm patients are shown in Table 1 and Table 2, respectively. The risk of bias assessment is reported in Table 3.

### 3.4. Main Findings

A total of 1613 endometrial cancer patients from 11 studies were tested for BRCA1/2 pathological variants [8,19,20,21,22,23,24,25,26,27,28]. Uterine serous carcinoma was diagnosed in 1129 patients. Of note, the diagnosis of uterine serous carcinoma was one of the inclusion criteria for 972 patients out of 1129. The notion of previous breast cancer was reported in 6 out of 10 studies; in particular, 47 out of 344 (13.7%) women with endometrial cancer had a personal history of breast malignancy. Only one paper reported the use of hormone therapy before endometrial cancer diagnosis [23]. BRCA1/2m were identified in 70 women with endometrial cancer out of 1613 (4.3%). Notably, BRCA1m represented 71.4% of BRCA pathological variants found in endometrial cancer patients (50/70).

A significant difference in the increased risk of endometrial cancer among BRCAm patients was found by Beiner [30] [SIR = 5.3 (*p* = 0.0011)], Saule [15] [SIR = 32.2 (95% CI, 11.5–116.4, *p* < 0.001)], and Thompson [29] [SIR = 2.65 (1.69–4.16, *p* < 0.001)]. However, two other studies confirmed this difference only for BRCA1m women [11,31]. Notably, Laitman et al. registered 14 cases of endometrial cancer among 2627 Jewish patients included in their study, assessing an increased overall rate of uterine cancer of almost 4-fold [SIR = 3.98 (95% CI, 2.174–6.673)] [11]. In a sub-analysis, this risk was significantly augmented for BRCA1 patients [SIR = 5.236 (95% CI, 2.659–9.382, *p* < 0.001)] but not for BRCA2 patients [SIR = 2.339 (95% CI, 0.743–5.642, *p* = 0.124)] [11]. Segev et al. confirmed the same results [BRCA1 = SIR 1.91 (95% CI, 1.06–3.19, *p* = 0.03), BRCA2 = SIR 1.75 (95% CI, 0.55–4.23, *p* = 0.2)] [31]. In contrast, Kitson et al. found no significant increased risk for endometrial cancer in the 2609 included in their study for both BRCA1/2m [SIR = 1.70 (95% CI, 0.74–3.33)] [37] as well as Goshen et al., even if they did not consider 2 of the 3 BRCA1 mutations examined in many other studies [20].

BRCA1/2 testing was generally performed by traditional Sanger sequencing. The next-generation sequencing (NGS) technique was otherwise adopted in 3 studies out of 24 [11,19,25].

The main findings of the included studies on the prevalence of BRCAm in patients with endometrial cancer are reported in Table 4.

A total of 209 BRCAm carriers from 14 studies diagnosed with BRCA1/2m were diagnosed with endometrial cancer [11,12,13,14,15,29,30,31,32,33,34,35,36,37]. A total of 9 studies out of 14 calculated the standardised incidence ratio (SIR) for the risk of uterine cancer in BRCAm women, dividing the total number of observed cases by the total number of expected cases [11,12,14,15,29,30,31,35,37]. Five authors found a statistical difference in the risk of endometrial cancer for BRCAm patients [11,15,29,30,31]. Nevertheless, two studies found a statistical difference only for BRCA1m women [11,31]. One study demonstrated an increased risk of developing aggressive and serous-like endometrial cancer, especially in BRCA1 mutation carriers [11]. The main findings of the included studies on the incidence of endometrial cancer in BRCAm patients are reported in Table 5.

## 4. Discussion

In this systematic review, we summarised all the studies that tested endometrial cancer patients for BRCAm and the incidence of endometrial cancer in BRCAm women. Controversial data have been found in the literature, and the correlation between BRCA mutations and uterine cancer is still debated. If a clear correlation were to be demonstrated, hysterectomy should systematically be added to bilateral salpingo-oophorectomy as a risk-reducing surgery. Shu et al. investigated the role of concomitant hysterectomy during RRSO in BRCAm patients to reduce the risk of uterine cancer and found that even though the overall risk for uterine cancer after RRSO was not increased, the risk for uterine serous cancer was increased in BRCA1m patients [12].

The Royal Australian and New Zealand College of Obstetricians and Gynaecologists classifies laparoscopic surgeries in 6 levels of complexity: 1. Diagnostic laparoscopy, 2. Salpingo-oophorectomy; 3. Laparoscopic-assisted vaginal hysterectomy; 4. Excision of ASRM stage 3 endometriosis and laparoscopic hysterectomy; 5. Laparoscopic myomectomy, excision of stage IV endometriosis; 6. excision of stage IV endometriosis necessitating bowel or urological resection, retroperitoneal lymphadenectomy, sacrocolpopexis [38]. Hence, adding hysterectomy to the risk-reducing procedures increases the degree of complexity of the procedure. This may reflect on various aspects. Firstly, a procedure that can nowadays be performed by virtually every gynaecologist may require a gynaecologist with more advanced surgical skills. Secondly, the addition of the hysterectomy to the bilateral salpingo-oophorectomy will increase operating room (OR) time and estimated blood loss, leading to an increase in direct costs linked to the length of hospital stay, medications required, OR time, etc. and indirect costs due to a loss of workdays related to the longer recovery period. Also, since the complication rate is usually related to the complexity of the procedure, a larger number of complications has to be expected. This aspect is of particular relevance when evaluating the risk-benefit balance of a prophylactic measure which is offered to women who are affected with a medical condition (BRCAm) but not with an illness. Usually, in experienced hands, the risk of iatrogenic lesions during a laparoscopic hysterectomy without additional complexity (e.g., endometriosis, fibroids, and adhesions) is very small. Furthermore, the long-term consequences of the hysterectomy need to be put into the equation, as well. A history of hysterectomy is associated with a slight increase in the risk of developing pelvic organ prolapse. In nulliparous women, who lack the most important risk factor for pelvic organ prolapse, namely having given vaginal birth, a history of hysterectomy increases the risk of developing pelvic organ prolapse by 60%. However, this increase in risk has a small clinical impact as it increases the risk from 12.7 to 20.5 per 100,000 risk years [39].

Another open question is how to process the endometrial lining of the uterus in case of risk-reducing surgery. Occult high grade serous ovarian cancer is identified in 6–17% of BRCAm carriers undergoing a risk-reducing salpingo-oophorectomy [40,41]. Nowadays, the pathological analysis of the tubes removed for risk-reducing surgery in BRCAm carriers is substantially different and more thorough as compared to when the tubes are removed secondary to another indication. In the first case, multiple sections of the ovaries and tubes should be performed to look for occult carcinoma using a specific protocol for patients at high risk of occult malignancy [40,41,42,43]. This labour-intense analysis increases the detection rate of occult ovarian or fallopian tube cancer in BRCAm women seven-fold [41]. Serous endometrial intraepithelial carcinoma (SEIC), a malignant lesion associated with p53 mutation in a background of atrophic endometrium, has been postulated to be a precursor of uterine serous carcinoma [44,45]. SEIC has been identified in 40–89% of patients diagnosed with serous endometrial cancer [46,47,48,49]. Furthermore, concordant genetic mutations have been demonstrated in both components of SEIC and serous endometrial cancer [50]. SEIC lesions may be focal and small, making the histological diagnosis and an extensive sampling of the uterus necessary to identify an invasive component [51]. This is of utmost importance since an extrauterine spread of disease in the absence of myometrial invasion has been described [51,52,53,54].

The advantages of a concomitant hysterectomy at the time of RRSO should also be taken into account for the management of menopause symptoms after surgery in BRCAm patients. Premature menopause in young women is one of the most important secondary effects of RRSO, leading to an increased risk of cardiovascular disease, bone mineral loss, and cognitive dysfunctions [55]. Many authors in the last decade have investigated the role and risks of hormone replacement therapy (HRT) in BRCAm women, and a distinction should be made between estrogen-only and combined estrogen and progestin HRT [56]. For women under 45 years of age who underwent RRSO, Kostopoulos et al. recorded a statistically significant protective effect on breast cancer of an estrogen-only HRT with an 18% risk reduction per year of treatment (95% CI, 0.69–0.97) [57]. On the contrary, a combined estrogen-progestin HRT confers a non-significative increase in breast cancer risk of 14% (95% CI 0.90–1.46) [57]. A striking pro-oncogenic role on mammary epithelial cells has been demonstrated for progesterone in a murine model [58].

In BRCA2m cancer-free women who do not want to undergo a prophylactic mastectomy, a chemoprevention strategy with Tamoxifen can be offered to reduce the incidence of breast cancer by 62% [59]. However, Tamoxifen is associated with a 2–3 fold increase in uterine malignancies [60,61,62]; therefore, hysterectomy at the time of the RRSO may be an option to avoid this risk. This option can also be considered for women with a BRCA1/2 mutation who underwent a mastectomy for breast cancer and who are taking Tamoxifen as adjuvant therapy.

## 5. Conclusions

This systemic review aims to provide clinicians with all recent data necessary for clear and exhaustive counselling about the benefits and risks of hysterectomy at the time of RRSO for BRCAm patients. As of now, data supporting the need to perform a hysterectomy at the time of RRSO are inconclusive, so a routine removal of the uterus should not be performed. However, this information should be discussed with the patient in order to offer a tailored approach.

Even if, to date, no guidelines recommend performing a hysterectomy as a risk-reducing procedure for HBOC syndrome, potential complications and costs of the surgical procedure (bleeding, infection, organ lesions, and vaginal cuff dehiscence) should be individually balanced with the potential increased risk of uterine cancer in this population and the reduced risks associated with an estrogen-only HRT.

## Figures and Tables

**Figure 1 jcm-11-03114-f001:**
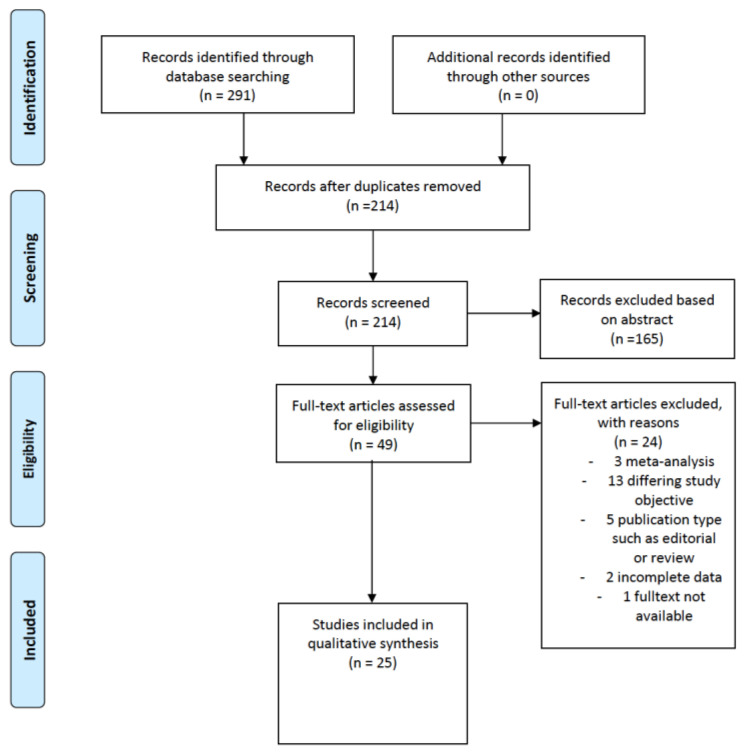
PRISMA flow chart of study identification.

**Table 1 jcm-11-03114-t001:** Characteristics of the included studies on the prevalence of BRCAm in patients with endometrial cancer.

Authors	Publication Year	Country	Time Period	Study Type	Study Group
Niederacher et al. [19]	1998	Germany	1980–1994	Retrospective case-control	EC
Goshen et al. [20]	2000	Canada	1996–2006	Retrospective multicenter cohort	USC
Levine et al. [21]	2001	Israel	1986–1998	Retrospective cohort	Jewish patients, EC
Lavie et al. [22]	2004	Israel	1999–2002	Retrospective multicenter cohort	USC
Biron-Shental et al. [23]	2006	Israel	1997–2003	Retrospective cohort	Jewish patients, USC
Barak et al. [8]	2010	Israel	1982–2008	Retrospective and prospective cohort	Jewish patients, EC
Bruchim et al. [24]	2010	Israel	1997–2007	Retrospective cohort	Jewish patients, USC
Pennington et al. [25]	2013	USA	NA	Retrospective cohort	USC
Mahdi et al. [26]	2015	USA	NA	Retrospective cohort	USC or ovarian serous carcinoma
Kadan et al. [27]	2018	Israel, Arabia	1993–2014	Retrospective multicenter cohort	USC
Vietri et al. [28]	2021	Italy	NA	NA	Hereditary EC (LS and HBOC)

BRCAm: breast cancer gene mutation, EC: endometrial cancer, USC: uterine serous carcinoma, LS: Lynch Syndrome, HBOC: Hereditary Breast and Ovarian Cancer syndrome, NA: not available.

**Table 2 jcm-11-03114-t002:** Characteristics of the included studies on the incidence of endometrial cancer in BRCAm patients.

Authors	Publication Year	Country	Time Period	Study Type	Study Group
Thompson et al. [29]	2002	Western Europe and North America	1960–2002	Retrospective multicenter cohort	BRCAm
Beiner et al. [30]	2007	North America, Europe and Israel	NA	Prospective multicenter cohort	BRCAm
Reitsma et al. [14]	2012	The Netherlands	1996–2012	Prospective cohort	BRCAm, RRSO
Segev et al. [31]	2013	Canada, Italy, USA, Austria, Poland, Norway	NA	Prospective multicenter case-control	BRCAm
Casey et al. [13]	2015	USA	1959–2013	Retrospective cohort	BRCAm with invasive gynecologic and/or peritoneal cancers
Segev et al. [32]	2015	North America, Europe and Israel	NA	Retrospective multicenter case-control	BRCAm
Shu et al. [12]	2016	USA, UK	1995–2011	Prospective multicenter cohort	BRCAm, RRSO
Zakhour et al. [33]	2016	USA	2000–2014	Prospective cohort	BRCAm, RRSO
Bogani et al. [34]	2017	Italy	2014–2017	Prospective cohort	BRCAm or significant family history of breast/ovarian cancer, RRSO ± hysterectomy
Lee et al. [35]	2017	Australia, New Zealand	NA	Prospective multicenter cohort	BRCAm
Minig et al. [36]	2018	Spain	2010–2017	Retrospective multicenter cohort	BRCAm, RRSO
Saule et al. [15]	2018	France	1996–2016	Prospective cohort	BRCAm, RRSO
Laitman et al. [11]	2019	Israel	1998–2016	Retrospective case-control	BRCAm
Kitson et al. [37]	2020	UK	1991–2017	Retrospective cohort	BRCAm

BRCAm: breast cancer gene 1/2 mutation, RRSO: risk-reducing salpingo-oophorectomy, NA: not available.

**Table 3 jcm-11-03114-t003:** Quality assessment of individual study.

Author	Bias due to Confounding	Bias in Selection of Partecipants	Bias Due to Missing Data	Bias in Classification of Interventions	Bias in Measurement of Outcomes	Bias in Selection of the Results	Overall
Niederacher et al., 1998 [19]	Moderate	Moderate	Moderate	Moderate	Moderate	Moderate	Moderate
Goshen et al., 2000 [20]	Serious	Serious	Serious	Moderate	Moderate	Serious	Serious
Levine et al., 2001 [21]	Moderate	Serious	Moderate	Moderate	Moderate	Low	Serious
Thompson et al., 2002 [29]	Moderate	Serious	Serious	Moderate	Moderate	Moderate	Serious
Lavie et al., 2004 [22]	Moderate	Serious	Moderate	Moderate	Low	Low	Serious
Biron-Shental et al., 2006 [23]	Serious	Moderate	Moderate	Moderate	Moderate	Low	Serious
Beiner et al., 2007 [30]	Low	Low	Low	Moderate	Moderate	Moderate	Moderate
Bruchim et al., 2010 [24]	Moderate	Low	Moderate	Moderate	Moderate	Serious	Serious
Barak et al., 2010 [8]	Moderate	Serious	Serious	Moderate	Low	Moderate	Serious
Reitsma et al., 2012 [14]	Low	Moderate	Low	Low	Moderate	Moderate	Moderate
Pennington et al., 2013 [25]	Moderate	Serious	Serious	Low	Low	Low	Serious
Segev et al., 2013 [31]	Low	Low	Low	Moderate	Moderate	Moderate	Moderate
Casey et al., 2015 [13]	Serious	Moderate	Moderate	Moderate	Moderate	Moderate	Serious
Mahdi et al., 2015 [26]	Moderate	Serious	Serious	Moderate	Low	Moderate	Serious
Segev et al., 2015 [32]	Moderate	Serious	Moderate	Moderate	Serious	Moderate	Serious
Zakhour et al., 2016 [33]	Moderate	Low	Moderate	Moderate	Moderate	Moderate	Moderate
Shu et al., 2016 [12]	Low	Moderate	Low	Moderate	Serious	Moderate	Serious
Lee et al., 2017 [35]	Moderate	Low	Low	Moderate	Serious	Moderate	Serious
Bogani et al., 2017 [34]	Low	Low	Low	Moderate	Moderate	Moderate	Moderate
Mining et al., 2018 [36]	Moderate	Serious	Serious	Moderate	Serious	Serious	Serious
Saule et al., 2018 [15]	Low	Moderate	Moderate	Moderate	Moderate	Moderate	Moderate
Laitman et al., 2018 [11]	Moderate	Serious	Serious	Moderate	Serious	Serious	Serious
Kadan et al., 2018 [27]	Moderate	Moderate	Moderate	Low	Moderate	Low	Moderate
Kitson et al., 2020 [37]	Moderate	Moderate	Moderate	Low	Moderate	Moderate	Moderate
Vietri et al., 2021 [28]	Low	Low	Low	Low	Low	Moderate	Moderate

**Table 4 jcm-11-03114-t004:** Main findings of the included studies on the prevalence of BRCAm in patients with endometrial cancer.

Author	Total Patients	Age, yr [Mean ± SD/Median (Range)] †	Genotyping			Total EC	EC Histopathology (n. of Patients; %)		USC	EC with Previous Breast Cancer	EC in Patients Using Tamoxifen	Positive Family History of Breast Cancer	Number of BRCA Mutated Patients			EC with BRCAm (%)
			Type	BRCA Mutation	Other Genes Tested		FIGO Stage	Grade					BRCA 1	BRCA 2	Tot	
Barak et al. [8]	289	63 ± 12	Traditional Sanger	BRCA1 (185delAG, 5383InsC, Tyr978X) BRCA2 (6174delT, 8765delAG)	-	289	In situ (2, 0.7%) I (234, 81%) II (24, 8%) III (25, 9%) IV (4, 1%)	In situ (2, 0.7%) I (168, 58%) II (50, 17%) III (69, 24%)	34	NA	NA	NA	4	1	5	1.7
Biron-Shental et al. [23]	22	72 (56–79)	Traditional Sanger	BRCA1 (185delAG, 5382insC) BRCA2 (6174delT)	-	22	I-II (9, 41%) III-IV (13; 59%)	NA	22	7	NA	7	3	3	6	22.7
Bruchim et al. [24]	31	72 (47–87)	Traditional Sanger	BRCA1 (185delAG, 5382insC) BRCA2 (617delT)	-	31	I-II (16, 52%) III-IV (15; 48%)	NA	31	7	6	5	4	4	8	25.8
Goshen et al. [20]	56	NA	Traditional Sanger	BRCA1 (185del AG, 5382insC, dup(ex13)) BRCA2 (6174delT)	-	56	I (27; 48%) II (6, 11%) III (13, 23%) IV (6, 11%) NA (4, 7%)	NA	56	6	NA	6	0	0	0	0
Kadan et al. [27]	64	66 ± 9.7 **	Traditional Sanger	BRCA1 (185delAG, 5382insC) BRCA2 (6174delT)	-	64	I (32; 50%) II (3, 5%) III (12, 19%) IV (16, 25%) NA (1, 1%)	NA	64	18	NA	NA	9	5	14	21.9
Lavie et al. [22]	20	72 (56–91)	Traditional Sanger	BRCA1 (185delAG, 5382insC) BRCA2 (6174delT)	-	20	I (NA, 30%) II (NA, 15%) III (NA; 40%) IV (NA, 15%)	NA	20	7	NA	7	4	0	4	20
Levine et al. [21]	199	66 ± 11	Traditional Sanger	BRCA1 (185delAG, 5382insC) BRCA2 (6174delT)	-	199	I (144; 72%) II (17, 9%) III (22, 11%) IV (14, 7%) NA (2, 1%)	1 (73, 37%) 2 (70, 35%) 3 (53, 27%) NA (3, 1%)	17	NA	NA	NA	1	2	3	1.5
Mahdi et al. [26]	241/628 *	68 (44–94)	NGS	NA	ABL1, AKT1, ALK, APC, ATM, BRAF, CDH1, CSF1R, CTNNB1, EGFR, ERBB2, ERBB4, FBXW7, FGFR1, FGFR2, FLT3, GNA11, GNAS, GNAQ, HNF1A, HRAS, IDH1, JAK2, JAK3, KDR (VEGFR2), KIT, KRAS, MET, MLH1, MPL, NOTCH1, NPM1, NRAS, PDGFR, PIK3CA, PTEN, PTPN11, RB1, RET, SMAD4, SMARCB1,SMO, STK11, TP53, VHL	628	NA	NA	628	NA	NA	NA	3	2	5	0.8
Niederacher et al. [19]	113	NA	Traditional Sanger	BRCA1-D17S855	TP53- AFM051, TCRD, ESR, D11S35, D16S511	113	I (69; 61%) II (18, 16%) III (15, 13%) IV (11, 10%)	1 (52, 46%) 2 (30, 27%) 3 (21, 19%) NA (10, 8%)	106	NA	NA	NA	13	0	13	11.5
Pennington et al. [25]	151	68	NGS	NA	APC ATM BAP1 BARD1 BMPR1A BRIP1 BUB1B CDH1 CDK4 CDKN2A CHEK2 KIT MLH1 MRE11A MSH2 (.EPCAM) MSH6 MUTYH NBN PALB2 PMS2 TEN (.KILLIN) RAD50 RAD51C RET SMAD4 STK11 TP53 VHL	151	I (61; 40%) II (16, 11%) III (34, 23%) IV (38, 25%) NA (2, 1%)	NA	151	2	NA	22	3	0	3	2
Vietri et al. [28]	40	35 (20–54) ***	NGS		MLH1, MSH2	40	NA		NA	NA	NA	NA	6	3	9	22.5

USC: uterine serous carcinoma; BRCAm breast cancer gene mutated patient, EC endometrial cancer, NA not available. † Data are expressed in mean or median as reported in each study. * Mahdi et al. included 5936 patients, of whom 5335 were affected by an ovarian serous carcinoma and 628 were affected by endometrial cancer, of which 241 were tested with NGS. ** Mean age of BRCA mutation carrier group. *** Mean age in EC patients.

**Table 5 jcm-11-03114-t005:** Main findings of the included studies on the incidence of endometrial cancer in BRCAm patients.

Author	Total Patients	Age, yr [Mean ± SD/Median (Range)] †	Total EC	EC Histopathology (n. of Patients; %)		EC with Previous Breast Cancer		EC in Patients Using Tamoxifen/Tot Patients Using Tamoxifen	History of Breast Cancer	Number of BRCA Mutated Patients			EC with BRCAm	Follow-Up [Mean/Median (Range), yr] or Women-Years (Median)	EC Risk in BRCAm (SIR [95% CI, *p*])
				FIGO Stage	Grade					BRCA 1 (BRCA1mEC/EC)	BRCA 2 (BRCA2mEC/EC)	Tot			
Beiner et al. [30]	857	54 (45–70)	6	I (4, 66%) II (1, 17%) NA (1, 17%)	1 (4, 66%) 2 (1 (17%) 3 (1, 17%)	5	0	4/226	551	619 (4/6)	236 (2/6)	857	6	3.3 (0.01–9.6)	5.3 (*p* = 0.0011)
Casey et al. [13]	101	NA	8	NA	1 (2, 25%) 2 (2, 25%) 3 (3, 38%) NA (1, 12%)	6	3	2/8	6/8	89 (7/8)	12 (1/8)	101	8	NA	NA
Kitson et al. [37]	2609	20 (20–32)	14	I (5, 36%) II (1, 7%) III (2, 14%) NA (6, 43%)	NA	NA	NA	NA	NA	1350 (7/14)	1259 (7/14)	2609	14	59,199 (23.8) women years	1.70 (0.74–3.33)
Laitman et al. [11]	2627	43 ± 7.7	14	NA	NA	7	5	2/178	1240	1746 (10/14)	1367 (4/14)	2627	14	32.744 † women years 20.468 †† women years	USC *** 14.29 (4.64–33.34, *p* < 0.001) Sarcoma *** 37.74 (10.28–96.62, *p* < 0.001) BRCA1 5.236 (2.659–9.382, *p* < 0.001) BRCA2 2.339 (0.743–5.642, *p* = 0.124)
Lee et al. [35]	828	43 (34–52)	5	I (3, 60%) II (2, 40%)	1 (2, 40%) 2 (1, 20%) 3 (2, 40%)	3	0	3/160	419	438 (3/5)	390 (2/5)	828	5	9.0	2.45 (95% CI: 0.80–5.72, *p* = 0.11) BRCA1 2.87 (95% CI 0.59–8.43, *p* = 0.18) BRCA2 2.01 (95% CI 0.24–7.30, *p* = 0.52)
Minig et al. [36]	359	49 ± 9.0	1	I (1, 100%)	NA	1	1	NA	225	223 (NA)	141 (NA)	359	1	2.4 (0.3–7.7)	NA
Reitsman et al. [14]	315	43 (30–71)	2	I (2, 100%)	NA	1	0	0/19	118	201 (1/2)	114 (1/2)	315	2	6 (0– 27)	2.13 (0.24–7.69; *p* = 0.27)
Saule et al. [15]	369	BRCA1 47 ± 1.3 BRCA 2 53 ± 6.8	2	IV (2, 100%)	NA	0	2	0/5	0	238 (2/2)	131 (0/2)	369	2	1779 woman-years	32.2 (11.5–116.4, *p* < 0.001)
Segev et al. [31]	4456	43	17	NA (17, 100%)	1 (5, 29%) 2 (1, 6%) NA (11, 65%)	10	1	8/697	1837	3536 (13/17)	920 (4/17)	4456	17	5.7	1.87, (1.13–2.94, *p* = 0.01) BRCA1 1.91 (1.06–3.19, *p* = 0.03) BRCA2 1.75 (0.55–4.23, *p* = 0.2)
Shu et al. [12]	1083	46 (41–53)	8	I (5, 63%) II (2, 25%) III (1, 12%)	NA	4	5	3/273	727	630 (5/8)	456 (3/8)	1083	8	5.1 (3.0–8.4)	1.9 (0.8–3.7, *p* = 0.09)
Thompson et al. [29]	7106 **	NA	47	NA	NA	NA	NA	NA	1928	2245 (11/11)	0	2245	11	NA	2.65 (1.69–4.16, *p* < 0.001)
Segev et al. [32]	14,621	52 (23–67) ****	83	NA	NA	46	NA	17/76	394/46	951 (62/83)	76 (21/83)	1027	83	NA	NA
Zakhour et al. [33]	257	46 (28–79)	1	II (1, 100%)	3 (1, 100%)	NA	0	NA	110	153	103 (1/1)	257	1	NA	NA
Bogani et al. [34]	85	47 ± 8.2	1	NA	NA	1	0	NA	60	32 (1/1)	25	57	1	1.5 ± 0.4	NA

BRCAm breast cancer gene mutated patient, EC endometrial cancer, USC uterine serous carcinoma, NA not available, BRCA1mEC breast cancer gene. One mutated patient with EC, BRCA2mEC breast cancer gene 2 mutated patient with EC. † women-years of follow up in BRCAm group. †† women-years of follow up in non-BRCA group. ** Thompson et al. included 11,847 patients, of whom 7106 were women. *** Risk in carriers group. **** Median age for cases.

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
