# Peer review of "Endometrial Cancer and BRCA Mutations: A Systematic Review"

_jcm, 2022, doi:10.3390/jcm11113114_

Round 1
Reviewer 1 Report
Dear Author,
I carefully read the Article “Endometrial Cancer and BRCA mutations: a systematic review”. It’s a well structured review and the topic is very interesting; however in the current literature there are similar meta-analysis evaluating the same data and reaching the same conclusions.
I have the following advices:
· at line 49 of page 2 you said that woman carrying BRCA mutations (BRCAm) are exposed to a higher risk of prostate cancer: this is surely a reference to the prevalence of BRCA mutation-associated cancer diseases in humans, but the way the sentence is articulated, you do not capture that meaning. please correct
· at Line 76 of page 2 there is the repetition of the word Pubmed,
· at line 68 of page 20, You affirm that in women with BRCA2 mutation who do not want to undergo a prophylactic mastectomy, a chemoprevention strategy with Tamoxifen can be offered, but Tamoxifene is associated with a 2-3 fold increase of uterine malignancies, so You say that hysterectomy at the time of the RRSO may be an option to avoid this risk. Here I would add that this option is possible also for women with BRCA1/2 mutation who underwent a mastectomy for breast cancer and who are taking Tamoxifene as adiuvant therapy.
Author Response
Dear Editor and dear reviewer,
Thank you for considering our manuscript for publication and for helping us in improving it.
Please find below the details on how we modified the manuscript according to your wise and useful suggestions.

Reviewer 2 Report
An interesting review, however some corrections should be performed; the authors did a review up to September 2021, but they should search for any recent published related studies, if any. Also some typing errors appear (eg, in methods section: the word Pubmed is repeated twice in the forst line,…) Kindly revise the whole manuscript. In figure 1 the full text articles excluded are 25 or 24 please revise.
Author Response

(The authors gave the same response as above.)

Round 2
Reviewer 1 Report
Dera Authors,
thank you for the edits performed, as requested.